# CoPhy: Counterfactual Learning of Physical Dynamics

**Fabien Baradel**[1]    **Natalia Neverova**[2]    **Julien Mille**[3]    **Greg Mori**[4]    **Christian Wolf**[1,5]

[1]Université Lyon, INSA Lyon, CNRS, LIRIS, Villeurbanne, France
[2]Facebook AI Research, Paris, France
[3]Laboratoire d'Informatique de l'Univ. de Tours, INSA Centre Val de Loire, Blois, France
[4]Simon Fraser University and Borealis AI, Vancouver, Canada
[5]Inria, Chroma group, CITI Laboratory, Villeurbanne, France

## Abstract

Understanding causes and effects in mechanical systems is an essential component of reasoning in the physical world. This work poses a new problem of counterfactual learning of object mechanics from visual input. We develop the CoPhy benchmark to assess the capacity of the state-of-the-art models for causal physical reasoning in a synthetic 3D environment and propose a model for learning the physical dynamics in a counterfactual setting. Having observed a mechanical experiment that involves, for example, a falling tower of blocks, a set of bouncing balls or colliding objects, we learn to predict how its outcome is affected by an arbitrary intervention on its initial conditions, such as displacing one of the objects in the scene. The alternative future is predicted given the altered past and a latent representation of the confounders learned by the model in an end-to-end fashion with no supervision of confounders. We compare against feedforward video prediction baselines and show how observing alternative experiences allows the network to capture latent physical properties of the environment, which results in significantly more accurate predictions at the level of super human performance.

## 1 Introduction

Reasoning is an essential ability of intelligent agents that enables them to understand complex relationships between observations, detect affordances, interpret knowledge and beliefs, and to leverage this understanding to anticipate future events and act accordingly. The capacity for observational discovery of *causal effects* in physical reality and making sense of fundamental physical concepts, such as *mass*, *velocity*, *friction*, etc., may be one of differentiating properties of human intelligence that ensures our ability to robustly *generalize* to new scenarios (Martin-Ordas et al., 2008).

One way to express causality is based on the concept of *counterfactual reasoning*, that deals with a problem containing an *if* statement, which is untrue or unrealized. Predicting the effect of the interventions based on the given observations without explicitly observing the effect of the intervention on data is a hard task and requires modeling of the causal relationships between the variable on which the intervention is performed and the variable whose alternative future should be predicted (Balke & Pearl, 1994). Using counterfactuals has been shown to be a way to perform reasoning over causal relationships between the variables of low dimensional spaces and has been an unexplored direction for high dimensional signals such as videos.

In this work, we develop the **Co**unterfactual **Phy**sics benchmark (**CoPhy**) and propose a framework for causal learning of dynamics in mechanical systems with multiple degrees of freedom, as illustrated in Fig. 1. For a number of scenarios, such as *tower of blocks falling*, *balls bouncing against walls* or *objects colliding*, we are given the starting frame $\mathbf{A} = X_0$ and a sequence of following frames $\mathbf{B} = X_{1:\tau}$, where $\tau$ covers the range of 6 sec. The observed sequences $\mathbf{B}$, conditioned on the initial state $\mathbf{A}$, are direct effects of the physical principles (such as *inertia*, *gravity* or *friction*) applied to the closed system, that cause the objects change their positions and 3D poses over time.

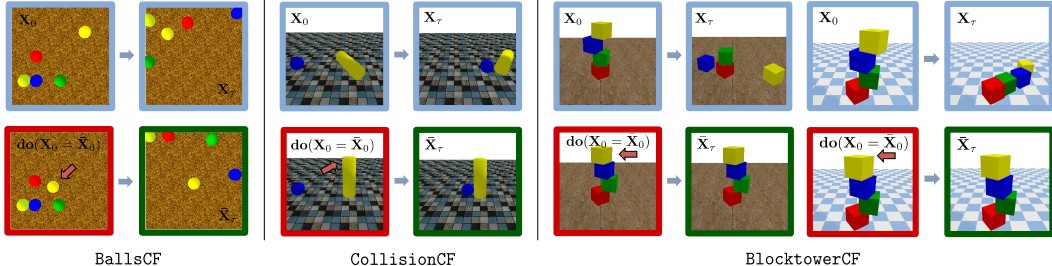

Figure 1: We train a model for performing counterfactual learning of physical dynamics. Given an observed frame $\mathbf{A} = X_0$ and a sequence of future frames $\mathbf{B} = X_{1:\tau}$, we ask how the outcome $\mathbf{B}$ would have changed if we changed $X_0$ to $\bar{X}_0$ by performing a *do*-intervention (e.g. changing the initial positions of objects in the scene).

The task is formulated as follows: having observed the tuple $(\mathbf{A}, \mathbf{B})$, we wish to predict positions and poses of all objects in the scene at time $t=\tau$, *if we had changed the initial frame $X_0$ by performing an intervention*. The intervention is formalized by the *do-operator* introduced by Pearl *et al.* (Pearl, 2009; Pearl & McKenzie, 2018) for dealing with causal inference (Spirtes, 2010). In our case, it implies modification of the variable $\mathbf{A}$ to $\mathbf{C}$, defined as $\mathbf{C} = \mathbf{do}(X_0=\bar{X}_0)$. Accordingly, for each experiment in the CoPhy benchmark, we provide pairs of original sequences $X_{0:\tau}$ and their modified counterparts $\bar{X}_{0:\tau}$ sharing the same values of all confounders.

We note the fundamental difference between this problem of *counterfactual future forecasting* and the conventional setup of *feedforward future forecasting*, like video prediction (Mathieu et al., 2016). The latter involves learning spatio-temporal regularities and thereby predicting future frames $X_{1...\tau}$ from one or several past frame(s) $X_0$ (the causal chain of this problem is shown in Fig. 2a). On the other hand, counterfactual forecasting benefits from *additional observations* in the form of the original outcome $X_{1:\tau}$ *before* the *do-operator*. This adds a *confounder variable $U$* into the causal chain (Fig. 2b), which provides information not observable in frame $X_0$. For instance, in the case of the CoPhy benchmark, observing the pair $(\mathbf{A}, \mathbf{B})$ might give us information on the masses, velocities or friction coefficients of the objects in the scene, which otherwise cannot be inferred from frame $\bar{X}_0$ alone. Therefore, predicting the alternative outcome after performing counterfactual intervention then involves using the estimate of the confounder $U$ together with the modified past $\mathbf{do}(X_0=\bar{X}_0)$.

Overall, we employ the idea of **counterfactual intervention in predictive models** and argue that counterfactual reasoning is an important step towards human-like reasoning and general intelligence. More specifically, key contributions of this work include:

• **a new task of counterfactual prediction** of physical dynamics from high-dimensional visual input, as a way to access capacity of intelligent agents for causal discovery;

• **a large-scale CoPhy benchmark** with three physical scenarios and 300k synthetic experiments including rendered sequences of frames, metadata (*object positions*, *angles*, *sizes*) and values of confounders (*masses*, *frictions*, *gravity*). This benchmark was specifically designed in **bias-free** fashion to make the counter-factual reasoning task challenging by optimizing the impact of the confounders on the outcome of the experiment. The dataset is publicly available[1].

• **a counterfactual neural model** predicting an alternative outcome of a physical experiment given an intervention, by estimating the latent representation of the confounders. The model outperforms state-of-the-art solutions implementing feedforward video prediction, successfully generalizes to unseen initial states and does not require supervision on the confounders. We provide extensive ablations on the effects of key design choices and compare results with human performance, that show that the task is hard for humans to solve. The code will be made publicly available.

## 2  RELATED WORK

This work is inspired by a significant number of prior studies from several subfields, including visual reasoning, learning intuitive physics and perceived causality.

---

[1]Project page: http://projet.liris.cnrs.fr/cophy

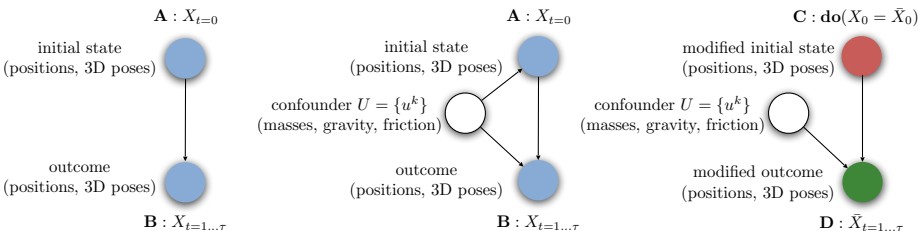

(a) feedforward future forecasting        (b) counterfactual future forecasting

Figure 2: The difference between conventional video prediction (a) and counterfactual video prediction (b). The causal graph of the latter includes a confounder variable, which passes information from the original outcome to the outcome after *do-intervention*. The initially observed sequence $(\mathbf{A}, \mathbf{B})$ (on the left) and the counterfactual sequence after the *do*-intervention (on the right).

**Visual reasoning.** The most recent works on visual reasoning approach the problem in the setting of visual question answering (Hu et al., 2017; Hudson & Manning, 2018; Johnson et al., 2017; Mao et al., 2019; Perez et al., 2018; Santoro et al., 2017), embodied AI (Wijmans et al., 2019), as well as learning intuitive physics (Lerer et al., 2016; Riochet et al., 2018). (Santoro et al., 2017) introduced Relation Networks (RN), a fully-differentiable trainable layer for reasoning in deep networks. Following the same trend, (Baradel et al., 2018) estimate object relations from semantically well defined entities using instance segmentation predictions for video understanding. (Santoro et al., 2018) build a challenging dataset for solving the problem of abstract reasoning on the visual domain with some tasks such as interpolation or extrapolation. External memory (Graves et al., 2016; Jaeger, 2016) extends known recurrent neural mechanisms by decoupling the size of a representation from the controller capacity and introduces the separation between long-term and short-term reasoning. (Reed et al., 2015) propose to learn analogies in a fully supervised way. Our work builds upon this literature and extends the idea of visual reasoning to the counterfactual setting.

**Intuitive physics.** Fundamental studies on cognitive psychology have shown that humans perform poorly when asked to reason about expected outcomes of a physics based event, demonstrating striking deviations from Newtonian physics in their intuitions (McCloskey & Kohl, 1983; McClooskey et al., 1980; 1983; Kubricht et al., 2017). The questions of approximating these mechanisms, learning from noisy observed and non-observed physical quantities (such as sizes or velocities vs masses or gravity), as well as justifying importance of explicit physical concepts vs cognitive constructs in intelligent agents have been raised and explored in recent works on deep learning (Wu et al., 2015). (Lerer et al., 2016; Groth et al., 2018) follow this direction by training networks to predict stability of block towers. (Ye et al., 2018) build an interpretable intuitive physical model from visual signals using full supervision on the physical properties of each object. On similar tasks, (Wu et al., 2017) propose to learn physics by interpreting and reconstructing the visual information stream leading to inverting physics or a graphical engine. (Zheng et al., 2018) propose to solve this task by first extracting a visual perception of the world state and then predict the future. (Battaglia et al., 2016) introduce a fully-differentiable network physics engine called Interaction Network (IN), which learns to predict physical systems such as gravitational ones, rigid body dynamics, and mass-spring systems. Similarly, (van Steenkiste et al., 2018) discover objects and their interactions in a unsupervised manner from a virtual environment. In (Veličković et al., 2018), attention and relational modules are combined on a graph structure. Recent approaches (Chang et al., 2017; Janner et al., 2019; Battaglia et al., 2018) based on Graph Convolution Networks (Kipf & Welling, 2017) have shown promising results on learning physics but are restricted to setups where physical properties need to be fully observable, which is not the case of our approach. The most similar to ours is work by (Ehrhardt et al., 2019) on *unsupervised* learning of intuitive physics from unpaired past experiences.

**Other physics benchmarks and simulators.** The main objective for the creation of our benchmark is (a) to focus specifically on evaluating capabilities of state of the art models for performing counterfactual reasoning, (b) to be unbiased in terms of distributions of parameters to be estimated and balanced with respect to possible outcomes, and (c) to have sufficient variety in terms of scenarios and latent physical characteristics of the scene that are not visually observed and therefore can act as confounders. To the best of our knowledge, none of existing intuitive physics benchmarks have these properties. IntPhys (Riochet et al., 2018) focuses on a high level task of estimating physical plausibility in a black box fashion and modeling out of distribution events at test time. CATER (Girdhar & Ramanan, 2019) introduces a video dataset requiring spatiotemporal understanding in

order to solve the tasks such as action recognition, compositional action recognition, and adversarial target tracking. Phyre (Bakhtin et al., 2019) is an environment for solving physics based puzzles, where achieving sample efficiency may implicitly require counterfactual reasoning, but this component is not explicitly evaluated, construction of parallel data with several alternative outcomes is not straightforward, and the trivial baseline performance levels are not easy to estimate. Adapting these benchmarks to counterfactual reasoning would require significant refactoring and changing the logic of the data sampling. CLEVRER (Yi et al., 2019) is a diagnostic video dataset for systematic evaluation of models on a wide range of reasoning tasks including counterfactual questions. They cast the task as a classification problem where the model has to choose between a set of possible answers, whereas our benchmark requires the predicting the dynamics of each object.

**Perceptual causality.** In the ML community, causal reasoning gained mainstream attention relatively recently (Lopez-Paz et al., 2017; Lopez-Paz & Oquab, 2017; Kocaoglu et al., 2018; Rojas-Carulla et al., 2018; Mooij et al., 2016; Schölkopf et al., 2012), due to limitations of statistical learning becoming increasingly apparent (Pearl, 2018; Lake et al., 2017). The concept of *perceived causality* has been however explored in cognitive psychology (Michotte, 1963), where human subjects have be shown to consistently report causal impressions not aligned with underlying physical principles of the events (Gerstenberg et al., 2015; Kubricht et al., 2017). Exploiting the *colliding objects* scenario as a standard testbed for these studies led to discovery of a number of cognitive biases, e.g. Motor Object Bias (i.e. false perceived association of object's velocity with its mass).

In this work, we bring the domains of visual reasoning, intuitive physics and perceived causality together in a single framework to tackle the new problem of counterfactual learning of physical dynamics. Following prior literature (Battaglia et al., 2013), we also compare counterfactual learning with human performance and expect that, similarly to learning intuitive vs Newtonian physics, modeling perceived vs true causality will get more attention from the ML community in the future.

## 3  CoPhy: Counterfactual Physics benchmark suite

In this paper we investigate visual reasoning problems involving a set of $K$ physical objects and their interactions, while considering a specific setting of **learning counterfactual prediction** with the objective of estimating objects' *alternative* 3D positions from images after *do-intervention*. We introduce the **Counterfactual Phy**sics benchmark suite (**CoPhy**) for counterfactual reasoning of physical dynamics from raw visual input. It is composed of three tasks based on three physical scenarios: `BlocktowerCF`, `BallsCF` and `CollisionCF`, defined similarly to existing state-of-the-art environments for learning intuitive physics: *Shape Stack* (Groth et al., 2018), *Bouncing balls* environment (Chang et al., 2017) and *Collision* (Ye et al., 2018) respectively. This was done to ensure natural continuity between the prior art in the field and the proposed counterfactual formulation.

Each scenario includes training and test samples, that we call *experiments*. Each experiment is represented by two sequences of $\tau$ synthetic RGB images (covering the time span of 6 sec at 5 fps):

• an **observed sequence** $X=\{X_0, \ldots, X_\tau\}$ demonstrates evolution of the dynamic system under the influence of laws of physics (gravity, friction, etc.), from its initial state $X_0$ to its final state $X_\tau$. For simplicity, we denote **A** the initial state $X_0$ and **B** the observed outcome $X_0, \ldots, X$;

• a **counterfactual sequence** $\bar{X}=\{\bar{X}_0, \ldots, \bar{X}_\tau\}$, where $\bar{X}_0$ (**C**) corresponds to the initial state $X_0$ after the *do-intervention*, and $\bar{X}_1, \ldots, \bar{X}_\tau$ (**D**) correspond to the counterfactual outcome.

A **do-intervention** is a *visually observable* change introduced to the initial physical setup $x_0$ (such as, for instance, object displacement or removal).

Finally, the physical world in each experiment is parameterized by a set of *visually unobservable* quantities, or **confounders** (such as object masses, friction coefficients, direction and magnitude of gravitational forces), that cannot be uniquely estimated from a single time step. Our dataset provides ground truth values of all confounders for evaluation purposes. However, we do not assume access to this information during training or inference, and do not encourage it.

Each of the three scenarios in the CoPhy benchmark is defined as follows (see Fig. 1 for illustrations).

**BlocktowerCF** — Each experiment involves $K=3$ or $K=4$ stacked cubes, which are initially at resting (but potentially unstable) positions. We define three different confounder variables:

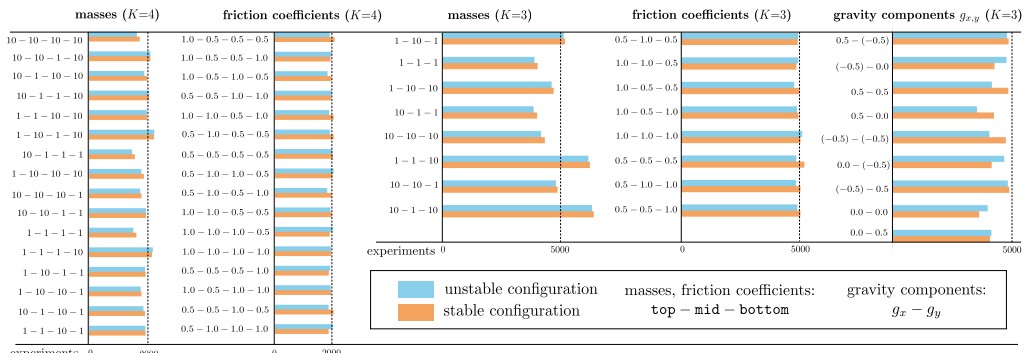

Figure 3: **Stability distribution for each confounder variable** for heights $K{=}3$ and $K{=}4$ of the `BlockTowerCF` task. Masses, friction cooefficients: 2 configurations per block, $2^K$ total; gravity: 3 configurations for each axis $\in\{x, y\}$, 9 total.

*masses*, $m\in\{1, 10\}$ and *friction coefficients*, $\mu\in\{0.5, 1\}$, for each block, as well as *gravity components* in $X$ and $Y$ direction, $g_{x,y}\in\{-1, 0, 1\}$. The *do-interventions* include block displacement or removal. This set contains 146k sample experiments corresponding to 73k different geometric block configurations.

**BallsCF** — Experiments show $K$ bouncing balls ($K{=}2...6$). Each ball has an initial random velocity. The confounder variables are the *masses*, $m\in\{1, 10\}$, and the *friction coefficients*, $\mu\in\{0.5, 1\}$, of each ball. There are two *do-operators*: block displacement or removal. There are in total 100k experiments corresponding to 50k different initial geometric configurations.

**CollisionCF** — This set is about moving objects colliding with static objects (balls or cylinders). The confounder variables are the *masses*, $m\in\{1, 10\}$, and the *friction coefficients*, $\mu\in\{0.5, 1\}$, of each object. The *do-interventions* are limited to object displacement. This scenario includes 40k experiments with 20k unique geometric object configurations.

Given this data, the problem can be formalized as follows. During *training*, we are given the quadruplets of visual observations $\mathbf{A}, \mathbf{B}, \mathbf{C}, \mathbf{D}$ (through sequences $X$ and $\bar{X}$, including GT object positions for supervision), but do *not* not have access to the values of the confounders. During *testing*, the objective is to reason on new visual data unobserved at training time and to predict the counterfactual outcome $\mathbf{D}$, having observed the first sequence $(\mathbf{A}, \mathbf{B})$ and the modified initial state $\mathbf{C}$ after the do-intervention, which is known.

The CoPhy benchmark is by construction **balanced and bias free** w.r.t. (1) global statistics of all confounder values within each scenario, (2) distribution of possible outcomes of each experiment over the whole set of possible confounder values (for a given do-intervention). We make sure that the data does not degenerate to simple regularities which are solvable by conventional methods predicting the future from the past. In particular, for each experimental setup, we enforce existence of at least two different confounder configurations resulting in significantly different object trajectories. This guarantees that *estimating the confounder variable is necessary for visual reasoning on this dataset*.

More specifically, we ensure that for each experiment the set of possible counterfactual outcomes is balanced w.r.t. (1) *tower stability* for `BlocktowerCF` and (2) *distribution of object trajectories* for `BallsCF` and `CollisionCF`. As a result, the `BlocktowerCF` set, for example, has $50 \pm 5\%$ of stable and unstable counterfactual configurations. The exact distribution of stable/unstable examples for each confounder in this scenario is shown in Fig. 3.

All images for this benchmark have been rendered into the visual space (RGB, depth and instance segmentation) at a resolution of $448 \times 448$ px with PyBullet (only RGB images are used in this work). We ensure diversity in visual appearance between experiments by rendering the pairs of sequences over a set of randomized backgrounds. The ground truth physical properties of each object (3D pose, 4D quaternion angles, velocities) are sampled at a higher frame rate (20 fps) and also stored. The training / validation / test split is defined as $0.7 : 0.2 : 0.1$ for each of the three scenarios.

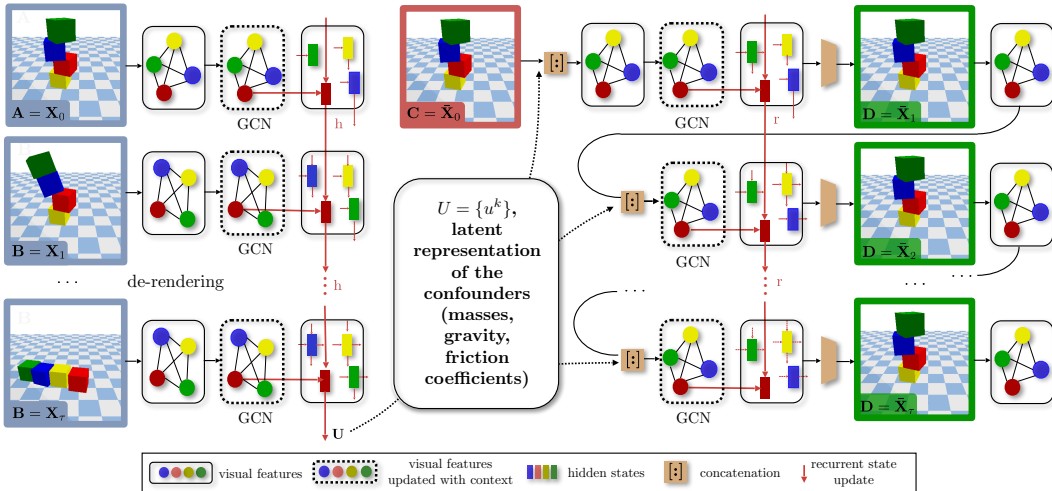

Figure 4: Our model learns counterfactual reasoning in a *weakly supervised* way: while we supervise the do-operator, we do *not* supervise the confounder variables (masses, frictions, gravity). Input images of the original past (**A**) and the original outcome (**B**) are de-rendered into latent representations which are converted into fully-connected attributed graphs. A Graph Network updates node features to augment them with contextual information, which is integrated temporally with a set of RNNs, one for each object, running over time. The last hidden RNN state is taken as an estimate of the confounder $U$. A second set of GCN+RNN predicts residual object positions (**D**) using the modified past (**C**) and the confounder representation $U$. For clarity we draw arrows for the red object only. *Not shown: stability prediction and gating*.

## 4 COUNTERFACTUAL LEARNING OF PHYSICS

The task as described in Section 3 requires reasoning from visual inputs.We propose a single neural model which can be trained end-to-end, as shown in Fig. 4. We address this problem by adding strong inductive biases to a deep neural network, structuring it in a way to favor counterfactual reasoning. More precisely, we add structure for (i) estimation of physical properties from images, (ii) modelling interactions between objects through graph convolutions (GCN), (iii) estimating latent representations of the confounder variables, and (iv) exploiting these representations for predictions of the output object positions. At this point we would like to stress again, that the representation of the confounders $U$ is latent and discovered from data without supervision.

### 4.1 UNSUPERVISED ESTIMATION OF THE CONFOUNDERS

While our method is capable of handling raw RGB frames as input, its internal reasoning is done on estimated representations in object-centric viewpoints. We train a convolutional neural network to detect the $K$ objects and their 3D position in the scene, denoted as $O=\{o^k\}, k=0... K-1$ where $o^k$ corresponds to the 3D position of object $k$. The de-rendering module is explained in the appendix.

Predicting the future of a given block $k$ requires modelling its interactions (through friction and collisions) with the other blocks in the scene, which we do with Graph Convolution Networks (GCN) (Kipf & Welling, 2017; Battaglia et al., 2018). The set of $K$ objects in the scene is represented as a graph $\mathcal{G}=(\mathcal{V}, \mathcal{E})$ where the nodes $V$ are associated to objects $\{o^k\}$, and the object interactions to edges $(o^k, o^j) \in \mathcal{E}$ in the fully-connected graph. Object embeddings $\{o^k\}$ are updated classically and as follows, resulting in new embeddings $\{\tilde{o}^k\}$:

$$e^k = \frac{1}{|\Omega_k|} \sum_{o^j \in \Omega_k} f(o^k, o^j), \quad e = \frac{1}{K} \sum_k e^k, \quad \tilde{o}^k = g(o^k, e^k, e), \tag{1}$$

where $\Omega_k$ is the set of neighboring objects of $o^k$. $f(.)$ and $g(.)$ are non-linear mappings (MLPs), and their inputs are by default concatenated. For simplicity, in what follows we will denote the update of an object $o^k$ with GCN given a graph with set of nodes and embeddings $O$ by $\tilde{o}^k = \mathbf{GCN}(o^k, O)$.

As mentioned above, we want to infer a latent representation $U$ of the *confounding* quantities for each object $k$ given the input sequences $X_{1:\tau}$ (the original past **A** and the original outcome **B**), without any supervision. This latent representation $U$ is trained end-to-end by optimizing the counterfactual prediction loss. To this end, we pass the updated object states $\tilde{o}^k$ through a recurrent network to model the temporal evolution of this representation. In particular, we run a dedicated RNN for each object, each object maintaining its own hidden state $h^k$:

$$h_t^k = \phi(\tilde{o}_t^k, h_{t-1}^k) \tag{2}$$

where we index objects and states with subscript $t$ indicating time, and $\phi$ is a gated recurrent unit (GRU) (gate equations have been omitted for simplicity). The recurrent network parameters are shared over objects $k$, which results in a model which is invariant to the number of objects present in the set. This allows to use do-operators which change the number of objects in the scene (removal). We set the latent representation of the confounders to be the set $U = \{u^k\}$, where $u^k \triangleq h_\tau^k$ is the temporally last hidden state of the recurrent network.

## 4.2 Trajectory prediction gated by stability

We predict the counterfactual outcome **D**, i.e. the 3D positions of all objects of the sequence $\bar{X}_{1:\tau}$, with a recurrent network, which takes into account the confounders $U$. We cast this problem as a sequential prediction task, at each time step $t$ predicting the residual position $\Delta_t^k$ w.r.t. to position $t-1$, i.e. the velocity vector. As in the rest of the model, this prediction is obtained object-wise, albeit with explicit modelling of the inter-object relationships through a graph network. More precisely,

$$\tilde{\bar{o}}_t^k = \mathbf{GCN}(\bar{o}_t^k, \{[\bar{o}_t^k : u^k]\}), \qquad r_t^k = \psi(\tilde{\bar{o}}_t^k, r_{t-1}^k), \qquad \Delta_t^k = \boldsymbol{W} r_t^k, \tag{3}$$

where $r_t^k$ is the hidden state of the GRU network denoted by $\psi$, and $\boldsymbol{W}$ is the weight matrix of a linear output layer. **GCN** is a graph convolutional network as described in eq. (1) and thereafter.

At each moment of time, each object can either remain stationary or move under the influence of external physical forces or by inertia. The first task for the model is therefore to detect which objects are moving (i.e. affected by the environment) and then estimate parameters of the motion if it occurs. This is aligned well with the concepts of **whether-causation** and **how-causation** defined in the field of *perceived causality* (Gerstenberg et al., 2015). In our work, the *whether-cause* is estimated in the form of a binary stability indicator $s_t^k$ described below (for each object, updated at each time step) that is then leveraged to gate the object position predictor (*how-cause* estimator):

$$\bar{o}_{t+1}^k = \bar{o}_t^k + \sigma\left(\frac{1 - s_t^k}{\lambda}\right)\Delta_t^k, \tag{4}$$

where $\sigma(.)$ is the sigmoid function and $\lambda$ is a sparsifying temperature term.

**Counterfactual estimation of stability** — estimation of object stability $s_t^k$ is a counterfactual problem, as stability depends on the physical properties, and therefore on the latent confounder representation $u_k$. We combine the confounders $U = \{u^k\}$ with the past after do-intervention (C), encoded in object states denoted as $\bar{O}_t = \{\bar{o}_t^k\}$ at time step $t=t$. In particular, for each node we concatenate its object features with its confounder representation and we update the resulting object state with a graph network to take into account inter-object relationships:

$$s_t'^k = \mathbf{GCN}([\bar{o}_t^k : u^k], \{[\bar{o}_t^k : u^k]\}), \qquad s_t^k = \boldsymbol{V} s_t'^k, \tag{5}$$

where $s_t^k$ corresponds to the logits of stability of object $k$ at time $t$ and $\boldsymbol{V}$ is the weight matrix of a linear layer (for simplifying notations we omit bias here and in the rest of the paper).

**Training** — The full counterfactual prediction model is trained end-to-end in graph space only (i.e. not including the de-rendering engine) with the following losses:

$$\mathcal{L}_{e2e} = \sum_{k=1}^{K} \mathcal{L}_{ce}(s_t^k, s_t^{k*}) + \sum_{t=0}^{\tau}\left[\sum_{k=1}^{K} \mathcal{L}_{mse}(\bar{o}_t^k, \bar{o}_t^{k*})\right]$$

where $\mathcal{L}_{ce}$ is the binary cross entropy loss between the stability prediction of object $k$ at time $t$ and its ground truth value (calculated by thresholding applied to ground truth speed vectors), and $\mathcal{L}_{mse}$ is mean squared error between the predicted positions $\bar{o}_t^k$ and GT positions $\bar{o}_t^{k*}$ in the 3D space. A detailed description of the overall architecture is given in the Appendix.

|  | Scenario | Top | Middle | Bottom | Mean | $\sigma$ |
|---|---|---|---|---|---|---|
| Human | Non-CF | 108.89 | 53. 15 | 13.67 | 58.57 | 33.61 |
|  | CF | 99.98 | 49.65 | 13.99 | 54.54 | 34.15 |
| Copying | $C{\to}D$ | 65.41 | 25.51 | 7.36 | 32.76 | N/A |
|  | $B{\to}D$ | 92.60 | 35.85 | 18.54 | 49.00 | N/A |
| **CoPhyNet** | Non-CF | 77.97 | 33.10 | 2.39 | 37.81 | N/A |
|  | **CF** | **57.10** | **23.26** | **4.40** | **28.25** | N/A |

Table 1: **Comparison with human performance in the `BlockTowerCF` scenario** obtained with AMT studies. We report 2D pixel error for each block, as well as global mean and variance $\sigma$ (reference resolution $448 \times 448$) on the test set with $K=3$ blocks.

| Train$\to$Test | Copy C | Copy B | IN | NPE | **CoPhyNet** | *IN sup.* |
|---|---|---|---|---|---|---|
| $3 \to 3$ | 0.470 | 0.601 | 0.318 | 0.331 | **0.294** | *0.296* |
| $3 \to 3$ † | 0.365 | 0.592 | 0.298 | 0.319 | **0.289** | *0.282* |
| $3 \to 4$ | 0.754 | 0.846 | 0.524 | 0.523 | **0.482** | *0.467* |
| $4 \to 4$ | 0.735 | 0.852 | 0.521 | 0.528 | **0.453** | *0.481* |
| $4 \to 4$ † | 0.597 | 0.861 | 0.480 | 0.476 | **0.423** | *0.464* |
| $4 \to 3$ | 0.480 | 0.618 | 0.342 | 0.350 | **0.301** | *0.297* |

Table 2: **`BlocktowerCF`:** MSE on 3D pose average over time. *IN sup.* methods in the last column exploit the ground truth confounder quantities as input and thus represent *a soft upper bound* (are not comparable). †Test confounder configurations not seen during training (50/50 split).

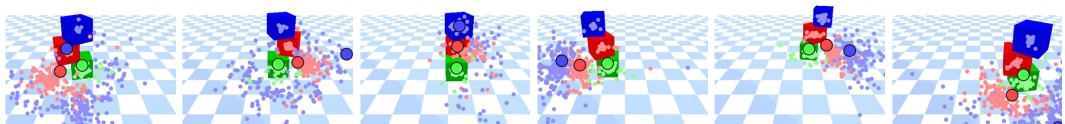

Figure 5: **Visual examples of human performance on the ill-posed task of feedforward, i.e. non-counterfactual, dynamic prediction from a single image (in the `BlockTower` scenario)**. The image shows the initial state **C**. Small dots correspond to human estimates of the objects' final positions. Larger circles indicate ground truth final positions of each block. We note that this task is ill-posed by construction, as the dynamics of each experiment is defined by physical properties of each block (e.g. masses) which cannot be observed from a single image.

## 5 EXPERIMENTS

**Training details.** All models were implemented in PyTorch. We used the Adam optimizer (Kingma & Ba, 2015) and a learning rate of 0.001. For training the de-rendering pipeline 200k frames were sampled for each of the three scenarios (see the appendix for more details).

**Human performance.** We empirically measured human performance in the `BlockTowerCF` scenario with crowdsourcing (Amazon Mechanical Turk/AMT). For this study, we have collected predictions from 100 participants, where each subject was given 20 assignments in both non-counterfactual (Fig. 5) and counterfactual (Fig. 6) settings. The human subjects were given 10 sec to click on the final positions of each block in the image **C** after the tower has fallen (or remained stable). The obtained quantitative results for both settings are reported in Table 1. We compare against copying baselines (i.e. predicting block positions in the frame **D** by either copying them from **C** or from **B**).

In conclusion, we observe that humans perform slightly better in the counterfactual setup after having observed the first dynamic sequence $(\mathbf{A}, \mathbf{B})$ together with **C** compared to the classical prediction where only **C** is shown. This behavior has also been previously observed in experiments on intuitive physics in cognitive psychology (Kubricht et al., 2017) that revealed poor human abilities to extrapolate physical dynamics from a single image. Similar human studies have also been conducted in (Battaglia et al., 2013) in a more simplistic setup of predicting the direction of falling, where the authors also reported that the task appeared to be challenging for human subjects.

The empirical results indicate that the participants decisions may have been however driven by simple inductive biases, e.g. "observed (in)stability in $(\mathbf{A}, \mathbf{B})$"$\to$"predict (in)stability in $(\mathbf{C}, \mathbf{D})$". The evidence for this approach is demonstrated qualitatively in Fig. 6: the variance in predictions after having observed a stable sequence is decreased (first row), after having observed a falling case – increased (second row). In all cases, human performance remains inferior w.r.t. the copying baselines.

The last part of Table 1 shows results of our model (denoted by **CoPhyNet** in the rest of the discussion) after projecting the estimated 3D positions of all objects back into the 2D image space. CoPhyNet significantly outperforms both human subjects and copying baselines.

**Performance and comparisons.** We evaluate the counterfactual prediction performance of the proposed CoPhyNet model against various baselines (shown in Tables 2-4 separately for each of the three scenarios of the CoPhy benchmark). The evaluated Network Physics Engine (NPE) (Chang et al., 2017) and Interaction Network (IN) (Battaglia et al., 2016), are both non-counterfactual baselines,

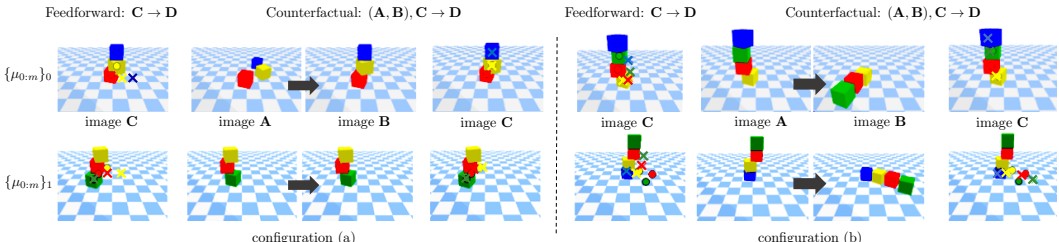

Figure 6: **Visual examples of human performance on the task of counterfactual dynamic prediction (in the `BlockTowerCF` scenario)**. Each participant has been shown both $(\mathbf{A}, \mathbf{B})$ and $\mathbf{C}$. Small dots correspond to human estimates of the objects' resting positions (outcome after do-intervention). Larger circles indicate the ground truth final positions. The images show state $\mathbf{C}$.

Figure 7: **Visual examples of the counterfactual predictions produced by CoPhyNet (in the `BlocktowerCF` scenario).** Circles denote GT position and crosses correspond to predictions.

that predict future block coordinates from past coordinates after do-intervention without taking the confounders into account. More details for IN and NPE are given in the appendix A.3. Our method consistently outperforms NPE and IN by a large margin in all scenarios. The CoPhyNet model also usually (but not always) outperforms the augmented variants of these methods that include the GT confounder quantities as input (a not comparable setting).

Fig. 7 illustrates several randomly sampled experimental setups and corresponding counterfactual predictions by the CoPhyNet model in the `BlocktowerCF` scenario.

**Generalization.** We evaluate the ability of the CoPhyNet model to generalize to new physical setups which were not observed in the training data. In Table 2 we show model performance on unseen confounder combinations and on unseen number of blocks in the `BlocktowerCF` scenario (lines marked with †). Our proposed solution generalizes well under unseen settings compared to other methods. In Table 3 we also demonstrate that our method outperforms the baselines by a large margin on unseen numbers of balls in the `BallsCF` setup. Finally, in the `CollisionCF` scenario (Table 4) we train on one type of moving objects and test on another type (spheres vs cylinders). In this case we also show that our method is able to generalize to the new object types even when it has not seen such a combination of <moving-object, static-object> before. Our method is able to estimate the object properties when an object is moving or initially stable.

**Impact of the confounder estimate.** Our model does not rely on any supervision of the confounders; we do, however, explore what effect supervision could have on performance, as shown in Table 6 (Middle). Adding the supervision increases the performance of the model for $K{=}3$ but the difference seems marginal (0.004 for $K{=}3$ and 0.020 for $K{=}4$). Directly feeding the confounder quantities as input leads to better performance, which is expected (but not comparable).

**Model architecture.** All design choices of CoPhyNet are ablated in Table 6 (Left) to fully illustrate the impact of each submodule. Estimating the stability once for the whole sequence $\mathbf{D}$ decreases the performance by 0.020 for $K{=}3$ and 0.018 for $K{=}4$ compared to predicting the stability per object at each time step. Replacing the GCN by a MLP (i.e. concatenating the object representation) hurts the performance of the overall system by increasing the MSE by 0.286 when tested in the $K{=}4$ setting. Finally we compare our approach against a single-step counterfactual prediction. Non-surprisingly, predicting the future autoregressively in a step-by-step fashion turns out to be more effective than predicting the whole sequence at once.

**Confounder estimation.** After training for predicting the target CF sequences, we evaluate the quality of the learned latent representation. In this experiment, we predict the confounder quantities of each object (mass, friction coefficient) from their latent representation by training a simple linear classifier, freezing the weights of the whole network. The obtained results are shown in Table 5. A

| Train→Test | Copy C | Copy B | IN | NPE | CoPhyNet | IN sup. |
|---|---|---|---|---|---|---|
| 4 → 2 | 7.271 | 3.267 | 5.060 | 4.989 | **2.307** | *2.109* |
| 4 → 3 | 6.820 | 2.865 | 4.895 | 4.901 | **1.990** | *1.886* |
| 4 → 4 | 6.538 | 2.688 | 4.785 | 4.821 | **1.978** | *2.069* |
| 4 → 5 | 6.221 | 2.568 | 4.732 | 4.817 | **1.958** | *2.346* |
| 4 → 6 | 6.045 | 2.488 | 4.661 | 4.668 | **1.899** | *2.564* |

Table 3: **`BallsCF`**: MSE on 2D pose average over time. *IN sup.* methods in the last column exploit the ground truth confounder quantities as input and thus is not directly comparable.

| Train→Test | Copy C | Copy B | IN | NPE | CoPhyNet | IN sup. |
|---|---|---|---|---|---|---|
| all→all | 4.370 | 0.665 | 0.701 | 0.697 | **0.173** | *0.332* |
| sphere→cylinder | 4.245 | 0.481 | 0.715 | 0.710 | **0.220** | *0.435* |
| cylinder→sphere | 4.571 | 0.932 | 0.720 | 0.699 | **0.152** | *0.586* |

Table 4: **`CollisionCF`**: MSE on 3D pose average over time. *IN sup.* methods in the last column exploit the ground truth confounder quantities as input and thus is not directly comparable (still showing inferior performance).

Table 5: **Ablations on `BlockTowerCF`: confounder prediction** (masses, friction coefficients) from the joint latent representation $U$. Metric: 4-way classification accuracy: Random=random classification.

| Method | 3 → 3 | 4 → 4 |
|---|---|---|
| Random | 25.0 | 25.0 |
| **CoPhyNet** | 65.7 | 68.9 |

| Method | 3 → 3 | 3 → 4 |
|---|---|---|
| Static gating | 0.305 | 0.496 |
| GCN replaced by MLP | 0.289 | 0.764 |
| Single-step prediction | 0.295 | 0.492 |
| **CoPhyNet** | **0.285** | **0.478** |

| Subset | Feedforward | | Counterfactual | |
|---|---|---|---|---|
| | confounders: input | – | confounders: supervision | – |
| K=4 | 0.248 | 0.349 | 0.281 | 0.285 |
| K=3 | 0.410 | 0.552 | 0.458 | 0.478 |

| Method | 3 → 3 | 3 → 4 |
|---|---|---|
| Copy C | 71.0 | 69.8 |
| Copy B | 69.9 | 68.5 |
| GCN(C) | 71.8 | 70.1 |
| **CoPhyNet** | **76.8** | **73.8** |

Table 6: **Ablation study on `BlockTowerCF`**: (Left) Impact of each component of our model (MSE on 3D pose average over time). (Middle) Impact of the confounder estimate (MSE on 3D pose average over time, validation set). Feedforward methods do not estimate the confounder, counterfactual methods do. We compare against soft upper bounds, which use the ground truth confounder as input or supervise its estimation. (Right) Stability prediction (accuracy per block). With the ground truth confounder values as input, graph convolutional networks (GCN(C)) reach a performance of $85.4$ and $77.3$ in the $3 \to 3$ and $3 \to 4$ settings respectively (*a soft upper bound*, not comparable).

prediction is correct if both the mass and the friction coefficients are correctly predicted. Our model outperforms the random baseline by a large margin suggesting that the confounder quantities are correctly encoded into the latent representation of each object during training.

**Stability prediction.** We studied the performance of the stability estimation module in the `BlockTowerCF` scenario and compared it to several baselines, as shown in Table 6 (Right). Our method predicts stability of each block from the confounder estimate $U$ and the frame $\mathbf{C}$. It outperforms the baselines estimating stability from a single input $\mathbf{C}$ or from the sequence $(\mathbf{A}, \mathbf{B})$ by a large margin, further indicating the efficiency of the confounder estimation and the complimentarity of this non-visual information w.r.t. the visual observation $\mathbf{C}$.

## 6 CONCLUSION

We formulated a new task of counterfactual reasoning for learning intuitive physics from images, developed a large-scale benchmark suite and proposed a practical approach for this problem. The task requires to predict *alternative* outcomes of a physical problem given the original past and outcome and an alternative past after do-intervention. Our challenging benchmarks cannot be solved by classical methods predicting by extrapolation, as the alternative future depends on confounder variables, which are unobservable from a single image of the alternative past. We train a neural model by supervising the do-operator, but not the confounders. Our experiments show that the CF setting outperforms conventional forecasting, and that the latent representation is related to the GT confounder quantities. We report human performance on this task, show its challenging nature and importance of CF reasoning.

We believe that counterfactual reasoning in high-dimensional spaces is a key component of AI and hope that our task will spawn new research in this area and thus contribute to bridging the gap between causal reasoning and deep learning. We also expect the benchmark to become a testbed in model based RL, which employs predictive models of an environment for learning agent behavior. Forward models are classically used in this context, but we conjecture that CF reasoning will contribute to disentangling representations and inferring causal relationships between different factors of variation.

**Acknowledgements.** This work was funded by grant Deepvision (ANR-15-CE23-0029, STPGP-479356-15), a joint French/Canadian call by ANR & NSERC.

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

## A  APPENDIX

### A.1  NEURAL DE-RENDERING

We train a convolutional neural network (CNN) to detect the $K$ blocks and estimate their positions in 3D, denoted as $O=\{o^k\}, k=0... K-1$ where $o^k$ corresponds to the position of object $k$. The network is inspired by classical region-based methods for object detection (He et al., 2017) and takes as an input an RGB image of resolution $224\times224$.

We define a *double convolution module* as a stack of a convolutional layer, batch normalization and ReLu activation, repeated two times. The resulting CNN includes three such modules with 64, 128 and 256 channels respectively, separated by $2 \times 2$ max pooling, which produces output feature maps of size $256\times56\times56$.

We design $K$ different heads by splitting the final feature maps channel-wise into $K$ feature maps and perform a double convolution module with 1 output channel. Each head outputs a feature map of size $1\times56\times56$ which is transformed into a vector to regress the object positions.

We first pre-train the de-rendering module alone without the model parts responsible for reasoning in the graph space. In particular, we de-render images $\hat{X}$ randomly sampled from **A**, **B**, **C** and **D** into its object representations $O=\{o^k\}$ and train with the following supervised loss:

$$\mathcal{L}_{\text{derender}} = \sum_{k=1}^{K} \mathcal{L}_{mse}(o^k, o^{k*}) \tag{6}$$

where $\hat{o}^{k*}$ are the ground truth 3D positions and $\mathcal{L}_{mse}$ corresponds to the mean square error. The rest of the model is trained end-to-end as described in section 4.2, paragraph "**Training**".

## A.2 OTHER ARCHITECTURES

Below are the descriptions of other networks used in our pipeline:

**f, g**  are implemented as MLPs with 4 and 2 layers respectively, with hidden layers of size 32 and ReLu activations.

$\phi, \psi$  are implemented as GRU modules with 2 layers and a hidden state of dimension 32.

**Confounders**  (mass and friction coefficients) are predicted with a single fully connected layer on top of the "confounder" representation of each object denoted $u^k$.

## A.3 FEEDFORWARD BASELINES

We compare our approach against two recent feedforward approaches, namely IN (Battaglia et al., 2016) and NPE (Chang et al., 2017). Both methods assume that GT object positions are available as input at training and test time, so they directly work on GT positions. They both predict the next position of each object using a Graph Convolution Network. IN is modeling object pairwise interaction between all objects in the scene, while NPE is taking into account only neighbouring objects for estimating the object interactions.

## A.4 TRAINING FROM ESTIMATED POSITIONS

In Table 7 we report the impact of the de-rendering module on performance. In particular, we compare performance of our model (CoPhyNet w/o GT, as described in the main part of the paper) with a version where we use ground-truth positions (CoPhyNet GT) for training. During training time, GT positions are fed to the main model. For testing, we do, however, use positions estimated by the de-rendering module in both versions.

We can see that training using the GT positions gives slightly better performance than training from estimated positions, which is expected.

| Train→Test | Copy C | Copy B | CoPhyNet GT | CoPhyNet w/o GT (=ours) |
|---|---|---|---|---|
| $3 \to 3$ | 0.470 | 0.601 | **0.294** | 0.309 |
| $3 \to 3$ † | 0.365 | 0.592 | **0.289** | 0.298 |
| $3 \to 4$ | 0.754 | 0.846 | **0.482** | 0.504 |

Table 7: `BlocktowerCF:` MSE on 3D pose averaged over time. We evaluate different types of training: from the ground-truth positions (GT) or from the estimated positions (w/o GT). †Test confounder configurations not seen during training (50/50 split).

