# OpenReview forum: "CoPhy: Counterfactual Learning of Physical Dynamics"
_ICLR.cc/2020/Conference — Accept (Spotlight)_

### Official Review · AnonReviewer3 · 2019-10-23
**Official Blind Review #3**

**Rating:** 6

**Review:**

Update: after revision, I have decided to keep the score unchanged.

Original comments:

In this paper, the authors proposed a new method to learn physical dynamics based on counterfactual reasoning.

1. As also summarized by the paper, over recent years, there has been increasing interest in the research community for the study of visual reasoning, intuitive physics and perceptual causality. This work provides an interesting framework that combines all the three domains together to solve the problem of learning physical dynamics. The experimental result also shows promise.

2. This paper also provides a nice work that bridges the gap between the counterfactual reasoning and deep learning community.

With all of this said, I think overall the paper is an interesting addition to the causal inference and deep learning literature.

**Experience Assessment:**

I do not know much about this area.

**Review Assessment: Checking Correctness Of Derivations And Theory:**

N/A

**Review Assessment: Checking Correctness Of Experiments:**

I assessed the sensibility of the experiments.

**Review Assessment: Thoroughness In Paper Reading:**

N/A

---

> ### Author Response · Authors · 2019-11-10
> **Response to Review 3**
>
> We are glad that R3 has appreciated this work as bridging the gap between deep learning and counterfactual reasoning / causal inference. We indeed hope that this will lead to increased communication and joint work between these communities.

---

### Official Review · AnonReviewer2 · 2019-10-23
**Official Blind Review #2**

**Rating:** 6

**Review:**

Summary of what the paper claims and contributes
---
This paper presents a new object forecasting task in the setting of "intuitive physics" that requires counterfactual reasoning and also presents method to perform this task. The task is to predict the alternative future trajectory of objects in 3D simulation given a video (A_frame,B_frames) of how the objects move in one situation and an alternative single frame (C_frame) that corresponds to modifying object position(s) at the first timestep of the input video. Due to unobserved confounding factors such as mass and friction that are not directly observable in either the video or the intervened frame, successful forecasting requires either implicitly or explicitly estimating these confounders. The motivation for this task is the hypothesis that counterfactual reasoning is a necessary component to forecast in general, unobserved situations. Prior work on explicit causal reasoning mainly concerns lower-dimensional problems, and prior work on high-dimensional "intuitive physics" has not evaluated the capability of models to perform counterfactual reasoning.

The paper proposes a neural network that learns to implicitly predict sufficient statistics of confounders of the situation. Training occurs in two stages. First, it is trained to estimate 3d positions of objects from images given pairs of (image, ground truth 3d positions). Then, it is trained to forecast future unobserved object positions with a graph convolutional network. Inference is performed by estimating the current object positions in all of the frames of the input (A_frame, B_frames), and then using the final representation of the input situation as the representation of confounders that is fed into the alternative-future prediction.

The experiments find:
1) Humans are not very good at these specific tasks, as evidenced by a simple position-copying baseline that outperforms humans. Given that the simple copying baseline is so performant, the claim in the abstract of "super human performance"  is also true for the copying baseline, and therefore that claim isn't very meaningful and should be removed.
2) The proposed approach outperforms position-copying and two non-counterfactual approaches that do not reason (explicitly or implicitly) about confounders,.
3) The approach attempts to show that confounder representations are learned as the main evidence that the model can perform counterfactual reasoning. However, several aspects of this experiment are unclear, making the claim difficult to evaluate.

Finally, although the abstract and the main text say that the task is to perform counterfactual forecasting in high-dimensional scenarios in a unsupervised way, the paper ends up using direct supervision of the object positions. With this knowledge, the problem is not that different than counterfactual forecasting in low-dimensional settings, as the positions of the objects at all frames corresponding to \mathbf{A}, \mathbf{B}, and \mathbf{C} could simply be extracted with this detector. Thus, through this supervision the approach seems to circumvent most of the challenges imposed by high-dimensional forecasting.

Evaluation
---

>Originality:
Are the tasks or methods new?
The task of counterfactual forecasting from high-dimensional observations is new. The proposed method extends prior work to the new task.

Is the work a novel combination of well-known techniques?
Yes.

Is it clear how this work differs from previous contributions?
Yes.

Is related work adequately cited?
To my knowledge, yes.

>Quality:
Is the submission technically sound?
Yes.

Are claims well supported by theoretical analysis or experimental results?
Some of the claims are supported, but the evidence for the counterfactual representation recovery claim is currently unclear. Additionally, the high-level motivation of counterfactual forecasting in high-dimensional settings is significantly undermined by the use of low-level supervision information that could feasibly be predicted at test time.

Is this a complete piece of work or work in progress?
It seems relatively complete.

Are the authors careful and honest about evaluating both the strengths and weaknesses of their work?
Weaknesses are not explicitly explored. It would be good to include discussion that illuminates or hypothesizes when and why the method fails, and how it could be made to perform better.

>Clarity:
Is the submission clearly written?
The submission is mostly clear, however there are some significant ambiguities in the experiments.

Is it well organized?
Yes.

Does it adequately inform the reader?
Mostly -- see ambiguities.

>Significance:
Are the results important?
It's currently unclear whether the results are significant, but it's clear the task is important (only when not supervised by object positions). However, justification with respect to other recent intuitive physics simulators is needed.

Are others (researchers or practitioners) likely to use the ideas or build on them?
Other researchers are likely to build on the task and data released.

Does the submission address a difficult task in a better way than previous work?
N/A (new task)

Does it advance the state of the art in a demonstrable way?
There is no SOTA on this new task.

Does it provide unique data, unique conclusions about existing data, or a unique theoretical or experimental approach?
Yes, one of the main strengths of the paper is the task and dataset that will be released. This experimental approach could be very useful for future research.

Additional feedback
---
The experiments to validate the cofounder estimation is unclear. Ambiguities:
0) In the text, the classification of mass and friction is presented as follows: "The obtained results are shown in Table 6 (middle)", yet the results are in Table 5. The impact of confounder supervision is described as follows "we do, however, explore what effect supervision could have on performance, as shown in Table 5", yet the results are in Table 6. These typos, when combined with the layout of Tables 5 and 6, makes it very difficult to interpret these results.
1) What is feedforward method? I can't find any description. Is this the IN or the NPE baseline, or something different? Why aren't both used? Without more details, its unclear that the proposed method actually learns confounder representations better than other methods.
2) What is the "random baseline level"? Is this just chance results from a uniform distribution as a baseline classifier?
3) What are the details of the classifier that predicts the mass and friction coefficients?

It is a bit odd to include comparisons to humans on a task on which the humans are outperformed by a very simple baseline. Either downplay the role of these comparisons, or remove these comparisons, or change the task such that humans cannot be outperformed by a simple baseline.

More details about the NPE and IN baselines are missing. Specifically, do they also leverage access to the ground-truth positions at training time? If so, how do they do this? This discussion is important for the paper to be self-contained.

Page 9 "The CophyNet model also consistently outperforms the augmented variants..." the outperformance isn't present in all cases (i.e. not in 4->2 or 4->3); change to "usually".

It's not clear if there is any stochasticity in the dynamics of the created data (which would be relevant to modeling and evaluation).

It wasn't clear until page 7 that the training leveraged GT positions in 3d space. The training objective (equation after Eq (5)) has an undefined o^*, which the reader must infer corresponds to GT, or read appendix A.1, to understand. The description of the method was introduced in terms of A, B, C, and D which are images or videos. To me, the fact that GT 3D positions are used seems like it was "buried" later in the paper, rather than described up-front alongside the introduction of A, B, C, and D.

The meaning of the colors, if any, in Fig 2 is unclear. If they are meant to illustrate observed information, perhaps grayscale shading could be used instead, as is standard in PGMs.

The latent representation in Fig 4 appear to emerge just from the red object's RNN, rather than all of the objects RNNs.

Tables 1 and 2 appear very far from their description in the text.

More discussion is needed that relates the tasks in to the tasks in Riochet2018 and Bakhtin2019. It's not clear why a new benchmark is needed without this discussion. Specifically, could the proposed method and evaluation be applied to these benchmark simulators? If not, why not? The answers to these questions seem to be the main motivation for the proposed tasks.

The simplest baseline would be to simply stack (A,B,C) as contextual input to a learned regressor of the future positions in D. Another simple baseline is to use the learned position estimator to predict all of the positions in (A,B,C) and use these positions as input to a learned regressor.

Use two backtick characters instead of the double-quote character for starting forward quotations.

Recommendation
---
Taken together, the novelty of the task and dataset, the discrepancy between problem motivation (high-dimensional counterfactual reasoning vs. uses low-dimensional ground-truth), and the partial clarity of the experimental results, leads me to conclude that although the proposed task and data are quite interesting and promising, the paper needs significant work to clarify its motivation and justification of its claims. I would rate the paper borderline, however it appears I'm forced to discretize to WA or WR. I choose WA because I am optimistic that the authors could address my doubts.


**Experience Assessment:**

I have published one or two papers in this area.

**Review Assessment: Checking Correctness Of Derivations And Theory:**

N/A

**Review Assessment: Checking Correctness Of Experiments:**

I assessed the sensibility of the experiments.

**Review Assessment: Thoroughness In Paper Reading:**

I read the paper at least twice and used my best judgement in assessing the paper.

---

> ### Author Response · Authors · 2019-11-10
> **Response to Review 2: Part 1**
>
> We are glad that R2 appreciates the novelty, importance and difficulty of the task, in particular that he estimates that "other researchers are likely to build on the task and data released". We want to thanks R2 for his valuable and detailed feedback. Below are our answers:
>
> * Solving the problem in high-dimensional space:
>
> Our input are high-dimensional images, but we provide supervision of GT positions during training time. We see this as an inductive bias, which currently makes it feasible to solve the problem, but we agree that at some point the goal should be to remove this. We think that the current state of the art in machine learning is not yet ready to tackle the problem directly.
>
> We would like to point out, that even our de-rendering module tackles a more difficult problem than the state of the art, which reasons on ground truth positions (done by the leading methods IN and NPE). Compared to this, our method goes one step further. Our object-centric approach still needs to address noise and incorrect estimations of the object positions, compared to existing literature.
>
> * Justification over recent intuitive physic simulators:
>
> The main objective of this study on the data construction side is creating a benchmark that is (a) focused specifically on evaluating capabilities of state of the art models for performing counterfactual reasoning, in a clean way, (b) unbiased in terms of distributions of parameters to be estimated and balanced with respect to possible outcomes (c) has sufficient variety in terms of scenarios and latent physical characteristics of the scene that are not visually observed and therefore can act as confounders. To be best of our knowledge, none of existing intuitive physics benchmarks have these properties. The IntPhys benchmark by Riochet et al. is focused on a high level task of estimating physical plausibility in a black box fashion and modeling out of distribution events at test time. Phyre by Bakhtin et al. is an environment for solving physics based puzzles, where achieving sample efficiency may implicitly require counterfactual reasoning, but this component is not explicitly evaluated, construction of parallel data with several alternative outcomes here is not straightforward, and the trivial baseline performance levels are not easy to estimate. Adapting these benchmarks for the task of counterfactual reasoning would require a significant refactoring and changing the logic of the data sampling.
> At the same time, we agree that the models discussed in our work could be adapted and inspire future work applied in these other settings as well.
>
> * Ambiguity on experiments
>
> ad 0) We updated the table references (confusion of references to tables 5 and 6) and apologize for this.
>
> ad 1) We consider a feedforward method an approach taking into account C only, so only the modified past and _NOT_ the original past (A) and outcome (B). This corresponds to classical non-counterfactual prediction of the future, which is an ill-posed problem since the confounder estimates are not available (unless explicitly fed in). We give a definition of feedforward future forecasting in Figure 2 (Left) and surrounding text.
>
> ad 2) The random baseline level in Table 5 corresponds to the chance results from a uniform distribution as a baseline classifier. This has been added to the revised version.
>
> ad 3) For the prediction of mass and friction coefficients, we use a dense layer on top of the "confounder" representation of each object denoted u^k. This description has been added to the appendix of the paper (other architectures section).
>
> * Human studies:
>
> We will downplay the comparison against human baselines, but we still think that it is interesting to know that humans are outperformed per simple baselines, since this is an experimental result per see, which to us was not obvious beforehand.

---

> > ### Author Response · Authors · 2019-11-10
> > **Response to Review 2: Part 2**
> >
> > * Further questions:
> >
> > Yes, NPE and IN are both feedforward methods which leverage access to the ground-truth positions at training and test time and which thus assume that the de-rendering step is solved. They reason on positions. We added a new paragraph to the appendix giving more details on IN and NPE.
> >
> > While CophyNet consistently outperforms the baselines in comparable settings, we agree that it does not always outperforms the augmented (i.e. ``cheating'') variants, we modified this statement in the revised paper.
> >
> > We did not introduce stochasticity into the created data. The prediction are fully deterministic given the object position and the confounder quantities of each object up to the inherent stochasticity of the physical simulator.
> >
> > As requested, in the revised paper We now introduce the supervision on GT object positions very early, when the sequences A,B,C,D are introduced at the beginning of section 3. We also introduce the symbol right after the equation of the training objective. This has been changed in the revised version of the paper.
> >
> > We also added experiments on GT object positions as input, i.e. without the  derendering step during training, to the appendix. The performance is slightly better when the model is fed GT positions during training, as expected.
> >
> > In Fig. 2 we use four different colors for differentiating the different states of the counterfactual process (blue for A and B (original past and outcome), red for C (modified/alternative past) and green for D (alternative outcome)). We are of course aware of standards in graphical models indicating the difference between observed (shown shaded) and latent (shown empty) variables. However, we think that the distinction between A,B,C,D is so central to our paper, that we chose these colors as a running thread used throughout the paper to make these differences clearer. They are used in figures 1, 2 and 4, so not only in the figure showing graphical model notation. We also respect GM notation in that all observed nodes are shaded (A,B,C,D are observed during training, the confounders are not).
> >
> > Unfortunately we will not have enough time during the rebuttal phase to produce the proposed baseline experiments (stacking A,B,C to produce D), but we agree that it makes sense to perform them, thank you for the remark. However, A+B alone is a sequence of non-neglectable length (20 instants), so stacking would very likely be a sub-optimal solution, akin to stacking in other sequence tasks like activity recognition or NLP, and as opposed to RNNs, 1D convolutions, or attention/transformers. On a related note, we experimented with replacing graph convolutional networks with stacking, which led to bad results, as expected.
> >
> > We replaced double quotes by back tics for forward quotes, as is LaTeX standard, agreed.

---

> > > ### Comment · AnonReviewer2 · 2019-11-14
> > > **Response to "Response to Review 2"**
> > >
> > > I am satisfied with your responses, but I would like to see them better-integrated into the paper so that future readers will not have the same doubts as me.
> > >
> > > - In the abstract, change "with no supervision" to "with no supervision of confounders" (so that it doesn't read like _only_ observations are available)
> > >
> > > - Please include more clarification in the paper about why a reader would want to use the proposed dataset instead of IntPhys or Phyre. This can probably be done easily by integrating the discussion from your response above. It should be clear from the paper, that, essentially: "If I want to solve/evaluate tasks X, simulators A and B may appear related, but they are missing Y which is important to be able to solve/evaluate X because of Z."

---

> > > > ### Author Response · Authors · 2019-11-14
> > > > **Re: Response to Review 2**
> > > >
> > > > We are pleased that the rebuttal addressed your comments. We agree that with some of our responses we targeted the reviewers more than the readers, and that these points should also be included in the paper. This now has been done.
> > > >
> > > > - The abstract has been changed as requested.
> > > >
> > > > - The related works section now contains a new paragraph ``Other physics benchmarks and simulators'' with a discussion which is heavily based on our rebuttal, and which should clearly distinguish the goals of CoPHy compared with IntPhys and Phyre.

---

### Official Review · AnonReviewer1 · 2019-10-23
**Official Blind Review #1**

**Rating:** 6

**Review:**

This paper studies counterfactual event prediction in physical simulation. The authors proposed a model that leverages object-centric scene representations and graph networks for modeling object interactions. The model also uses a recurrent network to encode and extract the confounder information for counterfactual prediction. In the experiments, the authors compared the proposed method, baselines, and human performance on pose estimation and counterfactual prediction.

I reviewed an earlier version of the paper at another venue. Compared with that, the current manuscript has improved a lot. It's studying an important problem. The model builds upon SOTA techniques such as GCN. Experiments are conducted on multiple physical events with multiple confounders. There are also rich ablation studies. The writing is clear and easy to follow. My recommendation is weak accept.

I think the paper can be improved by adding experiments on real data. The model involves 'de-rendering' which seems not easily generalizable to complex real scenes. Also, while the block tower scenario has been well studied, the discussion on ball and collision scenarios is quite limited. I encourage the authors to include more results on those datasets. The authors should also conduct human studies there, too.


**Experience Assessment:**

I have published in this field for several years.

**Review Assessment: Checking Correctness Of Derivations And Theory:**

I assessed the sensibility of the derivations and theory.

**Review Assessment: Checking Correctness Of Experiments:**

I assessed the sensibility of the experiments.

**Review Assessment: Thoroughness In Paper Reading:**

I read the paper at least twice and used my best judgement in assessing the paper.

---

> ### Author Response · Authors · 2019-11-10
> **Response to Review 1**
>
> We are glad that R1 appreciates the novelty and importance of the task. Below are our answers, we also revised and updated the paper itself.
>
> * Experiments on real data:
>
> Learning physics is a hard problem, which (up to our knowledge) has been addressed on simulated data only by the community up to now. In particular, we are not aware of any dataset where counterfactual learning can be performed on physics. Up to our knowledge we are the first to tackle this problem. We agree that an important task for the community will be to extend this type of problems to more general domains, but we think this is just not yet within the reach of our algorithms.
>
> * De-rendering module for complex real scenes:
>
> The de-rendering module can be seen as an object detection module, which should be easily generalizeable to more complex scenes, in particular when the backbone is changed to methods performing instance segmentation like Mask R-CNN. Recent works in action recognition have been using such object-centric approaches such as Actor-centric relation networks [1], Video as a Space-Time Graph [2] or Object Relational Networks [3]:
>
> [1] "Actor-Centric Relation Network", Sun, Shrivastava, Vondrick, Murphy, Sukthankar, Schmid, ECCV 2018
> [2] "Video as a Space-Time Graph", Wang, Gupta, ECCV 2018
> [3] "Object level visual Reasoning in Videos", Baradel, Neverova, Wolf, Mille, Mori, ECCV 2018
>
> * Ablation experiments on ball/collision scenarios.
>
> BlocktowerCF is arguably the most complex dataset of the three, so most of our ablation experiments are reserved on this dataset. That said, we do provide ablations on the two other datasets, showing the generalization of the model to unknown regimes: In Table 3 we provide results on BallsCF when the model is trained on N balls and tested on K balls where K!=N and we show that our proposed approach outperforms baselines. In Table 4 we provide results on CollisionCF when the model is trained with a certain type of moving object (e.g. sphere) and tested with another type of moving object (e.g. cylinder). We show that the model generalizes well compared to the baselines.
>
> * Human studies on ball/collision scenarios
>
> The human studies were meant to be complementary to the neural estimators. Given the short available time for the rebuttal we unfortunately cannot provide a human study for the two other scenarios (balls and collisions). We think that human studies on BlockTowerCF should provide indications about human performance on a counterfactual visual setup, in particular since the targeted subset is the most challenging one.

---

### Decision · Program_Chairs · 2019-12-19

**Decision:**

Accept (Spotlight)

**Comment:**

The reviewers are unanimous in their opinion that this paper offers a novel approach to learning naïve physics.  I concur.